# Nitric Oxide Release from Antimicrobial Peptide Hydrogels for Wound Healing

**DOI:** 10.3390/biom9010004

**Published:** 2018-12-21

**Authors:** Joana Durão, Nuno Vale, Salomé Gomes, Paula Gomes, Cristina C. Barrias, Luís Gales

**Affiliations:** 1i3S—Instituto de Investigação e Inovação em Saúde, Universidade do Porto, Rua Alfredo Allen, 208, 4200-135 Porto, Portugal; joana.oliveira@gmail.com (J.D.); sgomes@ibmc.up.pt (S.G.); ccbarrias@ineb.up.pt (C.C.B.); 2IBMC—Instituto de Biologia Molecular e Celular, Universidade do Porto, Rua Alfredo Allen, 208, 4200-135 Porto, Portugal; 3ICBAS—Instituto de Ciências Biomédicas Abel Salazar, Universidade do Porto, Rua de Jorge Viterbo Ferreira, 228, 4050-313 Porto, Portugal; 4Laboratório de Farmacologia, Departamento de Ciências do Medicamento, Faculdade de Farmácia, Universidade do Porto, Viterbo Ferreira 228, 4050-313 Porto, Portugal; 5IPATIMUP—Instituto de Patologia e Imunologia Molecular, Universidade do Porto, Rua Júlio Amaral de Carvalho, 45, 4200-135 Porto, Portugal; 6LAQV/REQUIMTE, Departamento de Química e Bioquímica, Faculdade de Ciências, Universidade do Porto, Rua do Campo Alegre, 687, 4169-007 Porto, Portugal; pgomes@fc.up.pt; 7INEB—Instituto de Engenharia Biomédica, Universidade do Porto, Rua Alfredo Allen, 208, 4200-135 Porto, Portugal

**Keywords:** Fmoc-Pexiganan, NO release, antimicrobial activity, collagen

## Abstract

Nitric oxide (NO) is an endogenously produced molecule that has been implicated in several wound healing mechanisms. Its topical delivery may improve healing in acute or chronic wounds. In this study an antimicrobial peptide was synthesized which self-assembled upon a pH shift, forming a hydrogel. The peptide was chemically functionalized to incorporate a NO-donor moiety on lysine residues. The extent of the reaction was measured by ninhydrin assay and the NO release rate was quantified via the Griess reaction method. The resulting compound was evaluated for its antimicrobial activity against *Escherichia coli*, and its effect on collagen production by fibroblasts was assessed. Time-kill curves point to an initial increase in bactericidal activity of the functionalized peptide, and collagen production by human dermal fibroblasts when incubated with the NO-functionalized peptide showed a dose-dependent increase in the presence of the NO donor within a range of 0–20 μM.

## 1. Introduction

Nitric oxide (NO) is an important biosignalling molecule with regulatory functions in the cardiovascular, immune, and central and peripheral nervous systems. The oxidation of L-arginine to L-citrulline and NO is catalyzed by NO synthase (NOS), an enzyme existing in three distinct isoforms: neuronal (nNOS), endothelial (eNOS), and inducible (iNOS) [1].

Different studies have suggested that nitric oxide synthesis is correlated to successful outcomes of wound healing. Shaffer et al. have reported a reduction in nitrite/nitrate concentration—the oxidation products of NO—in wound fluid upon administration of a competitive inhibitor of NOS-*S*-methyl isothiouronium to mice with a dorsal skin incision, and a concomitant decrease in collagen accumulation [2]. Yamasaki and co-workers have observed a 31% delay on time required for wound closure in iNOS knockout mice when compared with wildtype animals [3]. Endothelial NOS also plays a critical role in wound healing mechanisms. A study of excisional wound repair in eNOS knockout mice was observed to result in delayed wound closure time when compared with wildtype controls, as well as decreased incisional wound tensile strength [4]. It is thus reasonable to expect that NO-releasing materials might be a therapeutic option to improve wound healing [5,6,7,8,9,10].

Nitric oxide is a free radical and a highly reactive species, which greatly limits its action radius. Therefore, NO donor drugs are being developed. Nitric oxide-releasing drugs currently used in clinical practice mostly belong to the organic nitric category, which encompasses nitroglycerin (GTN) and isosorbide mononitrate (ISMN) (employed in the treatment of angina). However, these have been reported to lead to the development of tolerance with prolonged continuous use [11]. Other NO-releasing drugs used clinically include sodium nitroprusside (SNP), which is applied in hypertensive crises for an immediate reduction in blood pressure. Since this molecule is broken down by hemoglobin into cyanide, its administration encompasses the risk of cyanide poisoning [12].

Currently, NO donor drugs belonging to the diazeniumdiolates class are becoming known as promising therapeutic agents [13,14]. *N*-diazeniumdiolates, or NONOates (structural formula given in Scheme 1), are known to decompose spontaneously in solutions at physiological pH and temperature, giving rise to two molar equivalents (eq) of NO [15,16]. An extensive library of NONOates has been synthesized with half-lives that range from seconds to hours [17].

West et al. have developed NO-releasing hydrogels using different approaches, including the interesting exploitation of poly-L-lysine for the formation of NO adducts [18]. The authors first incorporated poly-L-lysine (degree of polymerization = 5) into poly(ethylene glycol) (PEG), which was then dissolved in water and reacted with NO gas to produce PEG-Lys5-NO. The PEG-Lys5-NO hydrogels were shown to reduce smooth muscle cell proliferation and platelet adhesion, which may be useful in the development of coatings to prevent thrombosis and restenosis. Several other NO-releasing materials have since been developed by adopting a similar strategy, that is, by the incorporation of primary amines into polymeric materials for NO adduct formation [19,20,21,22,23,24,25,26] such as poly(vinyl alcohol) (PVA) [21] and polyurethane modified by the incorporation of a peptide [19].

Short peptides have been used to produce microporous materials suitable for physical adsorption of small gas molecules [27]. The adsorption of Xe [28], CO_4_ and H_2_ [29], Ar [30], and O_2_ and N_2_ [31] have already been investigated. Usually the gas uptake and release from these materials are very fast [32], making them unsuitable for NO delivery applications. On the other hand, reports on the successful production of NO nucleophile complexes derived from primary amines have encouraged us to envisage the formation of a NO-releasing wound dressing derived from a self-assembling peptide hydrogel. Among the countless known antimicrobial peptides [33], we selected MSI-78, also known as pexiganan, because of its particularly hydrophobic and aromatic-rich peptide sequence, GIGKFLKKAKKFGKAFVKILKK, which confers a greater likelihood for self-assembly [34,35,36,37,38,39,40,41,42,43,44,45,46,47]. Pexiganan has a broad spectrum of antimicrobial activity against Gram-positive and Gram-negative aerobes and anaerobes, and is thought to act by disturbing the permeability of the cell membrane or cell wall [48]. 

Wound dressings are a valuable part of chronic wound treatment and should be designed to remove exudates, prevent infection, and foster healing. A recently published review summarized the effects on ulcer healing of hydrogels with alternative wound dressings [48]. The authors concluded that there is some evidence of the greater healing capacity of hydrogels when compared to basic wound contact dressings, despite concerns about possible bacterial growth due to the moist environment [49,50,51].

In view of the above, it is reasonable to assume that a hydrogel composed of antimicrobial peptides would provide both the complementary effects of an antimicrobial hydrogel—providing moisture to the wound bed and allowing nutrients and gases to diffuse through, while reducing the risk of infection—and the ability to locally release exogenous NO in a controlled manner, improving wound healing.

## 2. Materials and Methods

### 2.1. Peptide Synthesis

Fmoc-Pexiganan (Fmoc-PXG) and Pexiganan (PXG) were synthesized via microwave-assisted solid phase peptide synthesis (MW-SPPS), on a CEM Liberty1 instrument (CEM Corporation, Mathews, NC, USA), employing the Fmoc/^t^Bu approach [52,53]. Briefly, Fmoc-Rink-4-methylbenzhydrylamine (MBHA) resin was preconditioned for 15 min in *N*,*N*-dimethylformamide (DMF), and then transferred into the MW-reaction vessel. The initial Fmoc deprotection step was carried out using 20% piperidine in DMF containing 0.1 M of 1-hydroxybenzotriazole (HOBt), in two MW irradiation pulses—30 s at 24 W plus 3 min at 28 W—with in both cases the temperature being no higher than 75 °C. The C-terminal amino acid was then coupled to the deprotected Rink amide resin using 5 eq of the Fmoc-protected amino acid in DMF (0.2 M), 5 eq of 0.5 M HBTU/HOBt in DMF, and 10 eq of 2 M *N*-ethyl-*N*,*N*-diisopropylamine (DIPEA) in *N*-methylpyrrolidone (NMP). The coupling step was carried out for 5 min at 35 W MW irradiation, with a maximum temperature of 75 °C. The remaining amino acids were sequentially coupled in the C→N direction by means of similar deprotection and coupling cycles. Double-coupling was employed when coupling lysines in the sequence, which were incorporated as Fmoc-Lys(Boc)-OH. Following completion of the sequence assembly, the peptide was released from the resin with concomitant removal of side-chain protecting groups by a 2 h acidolysis at room temperature using a trifluoroacetic acid (TFA)-based cocktail containing triisopropysilane (TIS) and water as scavengers (95:2.5:2.5 *v/v/v*). Peptide purification was accomplished using a preparative medium-pressure liquid chromatography (MPLC) column packed with octadecyl carbon chain (C18)-bonded silica as the stationary phase. The purified products were analyzed by reverse phase high-pressure liquid chromatography (RP-HPLC) and electrospray ionization mass spectrometry (ESI-MS, Finnigan Surveyor LCQ DECA XP MAX).

Purified peptide solutions were frozen and subsequently lyophilized, and the resulting peptide powders kept at −20 °C until used. PXG was produced by removal of the Fmoc group from the N-terminal amino acid prior to cleavage, whereas Fmoc-PXG was released from the resin without having carried out such an N-terminal deprotection step.

### 2.2. Gelation of Antimicrobial Peptides

Both peptides, PXG and Fmoc-PXG, were dissolved in ultrapure water (MilliQ) (which had been previously filtered through a 0.22 μM pore membrane filter) and an aqueous sodium hydroxide 0.1 M solution was added to a final peptide concentration of 2.5% (*w/v*). Hydrogel formation was confirmed by inversion of the flask.

### 2.3. Formation of the Nitric Oxide Nucleophile Complex

The reaction procedure used to produce *N*-diazeniumdiolates from primary amines was adapted from those published in previous reports [18,19]. The peptide was dissolved in ultrapure water in a glass vial, and an aliquot was collected and stored at 4 °C as a control solution for future experiments. The glass vial with the remaining peptide solution was then placed in a reaction vessel with a magnetic stir bar to allow constant mixing of the solution, and the reactor sealed. The reactor was purged with N_2_ to remove O_2_. Afterwards, the reaction vessel was filled with NO gas (50% in N_2_) at around 2.5 bar and allowed to react for approximately 18 h under constant stirring.

Following a secure evacuation of the NO charged atmosphere, the solution was withdrawn from the reaction vessel and samples were collected for characterization and evaluation of the extent of conversion of free amines. The remaining solution was frozen at −80 °C overnight and freeze-dried.

### 2.4. Extent of Reaction: Analysis of Nitric Oxide Nucleophile Complex Formation

The ninhydrin reagent was developed for the quantitative determination of amino acids through its reaction with primary amine groups, which produced the colored ninhydrin chromophore named Ruhemann’s purple (λ_max_ = 570 nm; ε = 22 000 M^−1^·cm^−1^) [54,55]. This assay was used as an indirect method to quantify the extent of conversion of free amines, from lysine side-chains to NO-nucleophile complexes, which has been described as the established methodology for primary amine functionalization with NONOates.

Ninhydrin solution was made fresh for every experiment and quantities were adjusted according to the volume required. Briefly, for the preparation of a 10 mL solution, 30 mg of hydrindantin and 200 mg of ninhydrin were dissolved in 7.5 mL of dimethylsulfoxide (DMSO). Immediately prior to analysis, 2.5 mL of a 4 M sodium acetate buffer solution at pH 5.2 was added. Unknown samples (0.5 mL) and ninhydrin solution (0.5 mL) were added to a screw-capped test tube and heated in a boiling water bath for 15 min. After cooling the samples in an ice bath to stop the reaction, 2.5 mL of a 50% ethanol solution was added and vigorously mixed. Absorbance was monitored at 570 nm (Shimadzu UV-2401 PC, Shimadzu Corporation, Kyoto, Japan).

A standard curve was obtained by reacting glycine solutions, prepared in ultrapure water, with the ninhydrin reagent as previously described. The obtained standard curve was linear for glycine concentrations ranging from 10 to 200 μM. From linear regression it was possible to quantify the free amines. A dilution of the sample was always in place to allow the amine quantification values to remain within the linear region of the standard curve and a control (pre-reaction) sample was used for calibration.

### 2.5. Kinetics of Nitric Oxide Release

The NO release was measured using a Griess assay, which measures NO indirectly by quantifying nitrite, NO_2_^−^. The reaction of nitrite with sulfanilamide or sulfanilic acid forms a diazonium salt intermediate that then reacts with *N*-(1-napthyl) ethylenediamine to form an azo dye with a peak absorbance at 548 nm [56]. Under acidic conditions, NO released can be measured by spectroscopically monitoring the solution at 548 nm.

Griess reagent was prepared by mixing equal volumes of a solution of *N*-(1-naphthyl) ethylenediamine dihydrochloride (1 mg/mL) and a sulfanilic acid (10 mg/mL) solution in 5% phosphoric acid. Reaction mixtures were prepared using the proportions 100 μL of Griess reagent, 300 μL of the nitrite-containing sample, and 2.6 mL of deionized water. Reaction solutions were allowed to react for 30 min in a light-protected environment at room temperature. Each solution was then pipetted into a 1 cm path length cuvette and absorbance was monitored at 548 nm (Shimadzu UV-2401 PC).

### 2.6. Antimicrobial Activity (Time-Kill Curves)

The bactericidal action of the functionalized peptide against *Escherichia coli* was determined by generating time-kill curves. Glass tubes containing different concentrations of either the control, the functionalized peptide, or a blank solution were inoculated with a suspension of *E. coli* at a final concentration of approximately 1 × 10^6^ colony-forming unit (CFU)/mL. The tubes were subsequently incubated at 37 °C and viable counts were performed at different time points (0, 0.5, 1, 2, 3 and 5 h) after peptide addition. To perform colony counts, aliquots of culture broth were taken after careful homogenization at the predefined time points, serially diluted in sterile phosphate-buffered saline (PBS), and spread in duplicates over Nutrient Agar plates. These were then incubated overnight at 37 °C and colonies were counted.

### 2.7. In Vitro Assessment of Collagen Expression

Human Dermal Neonatal Fibroblasts (ZenBio, Inc., Research Triangle Park, North Carolina, USA) were grown in tissue culture flasks at 37 °C in a 5% CO_2_ controlled atmosphere, in Dulbecco’s modified Eagle’s medium (DMEM) (Gibco/BRL, Gaithersburg, MD, USA) supplemented with 10% (*v/v*) fetal bovine serum (Gibco, Waltham, Massachusetts, USA). Subculturing was performed by tripsinizing cultures with 0.25% Trypsin (Sigma-Aldrich, Darmstadt, Germany) and 0.05% ethylenediaminetetraacetic acid (EDTA) (Sigma-Aldrich). Experiments were performed at passage 15.

Fibroblasts were seeded at 2 × 10^5^ cells/well in four 6-well culture plates and incubated for approximately 48 h at 37 °C and 5% CO_2_. Upon reaching confluence, the cells were subjected to a serum starvation period of 6 h by replacing the culture medium with DMEM without fetal bovine serum (FBS). Following this, the culture medium was supplemented with 500 µM ascorbic acid (2-phospho-L-ascorbic acid trisodium salt) and increasing concentrations of Fmoc-PXG/NO and Fmoc-PXG (0, 5, 10, 20, 50, and 100 μM). The culture plates were then incubated at 37 °C in a 5% CO_2_ controlled atmosphere for a period of 23 h.

### 2.8. Collagen Quantification

Collagen was assessed using a Sircol assay (Biocolor, Belfast, United Kingdom) according to instructions provided by the manufacturer, with the exception of the isolation and concentration step which was replaced with an improved procedure [57]; the negative control was solvent. Accordingly, 1 mL of Sircol reagent was added to 100 μL of sample and left reacting in a shaker for 30 min at room temperature. The collagen-dye complex precipitate was deposited at the bottom of the microcentrifuge tube by centrifugation and the solution drained. The precipitate was then carefully washed to remove unbound dye from the surface of the pellet as well as from the interior surface of the microcentrifuge tube. After solution centrifugation and drainage, the precipitate was dissolved in a 250 μL alkali reagent by applying vigorous mixing. A 200 μL volume of each sample was transferred to individual wells on a 96-well microplate and absorbance was read at 540 nm in a microplate reader (Synergy MX, Biotek, Winooski, Vermont, USA). Collagen concentrations were determined using standards and a calibration curve. When measuring collagen deposited onto cell culture plastic surfaces and arising from the endogenously produced extracellular matrix, an extra step was required involving overnight incubation of the sample in an acid-pepsin solution at 4 °C. Following this step, the previously described procedure was carried out.

### 2.9. DsDNA Quantification

DNA quantification was performed using a Quant-iT PicoGreen dsDNA kit (Molecular Probes, Eugene, Oregon, USA). Upon binding of the PicoGReen reagent to dsDNA an increased fluorescence was observed, which could be correlated to the number of cells present in the sample [58] Twenty-two hours following the addition of components to the cells, a PicoGreen assay was performed according to the manufacturer’s instructions. Lysis was accomplished by treatment w.ith Triton X-100 1% after overnight freezing of cell plates. Lysed cell solutions were then added to each well in triplicate along with standard dsDNA solution to a 96-well microplate. PicoGreen working solution was introduced to each well, followed by incubation in a light-protected environment at room temperature for 5 min. Fluorescence signals were detected using a fluorescent microplate reader (Synergy MX, Biotek) at 480 nm (excitation) and 520 nm (emission).

## 3. Results and Discussion

### 3.1. Peptide Syntheseis and Gelatination

The resulting purified Fmoc-PXG and PXG peptides were characterized by RP-HPLC and ESI-MS. Peptides were obtained with high purity (>95%) as measured by RP-HPLC. When sodium hydroxide solution was added to Fmoc-PXG, an immediate phase transition was observed which resulted in a translucent hydrogel. The self-supporting ability of the hydrogel was verified by simply inverting the container and observing if there was any collapse of the formed hydrogel. The peptide lacking the Fmoc aromatic group, PXG, did not suffer any transition or visible aggregation in the same conditions and remained a clear solution at all times. Both samples were photographed and are displayed in Figure 1.

### 3.2. Incorporation of Nitric Oxide Donor Moiety and Kinetics of Nitric Oxide Release

The reaction of Fmoc-PXG with NO was performed as previously described and aliquots of the resulting solution were collected in triplicate alongside the control solution (Fmoc-PXG) and the blank solution (ultrapure water). The samples were assessed for free amines via a ninhydrin assay and there was good reproducibility among the replicates. Under constant experimental conditions (reactor volume and NO pressure), the efficiency in the reaction of primary amines decreased with the increase in the initial amount of peptide, ranging from a conversion of 60% obtained with 30 μM Fmoc-PXG solution down to 14% obtained with 400 μM Fmoc-PXG solution. NO released was quantified via Griess reaction assay. Blank (ultrapure water) and control (Fmoc-PXG) solutions were processed in the same way as the Fmoc-PXG/NO sample. Absorbance was monitored at different time-points at 540 nm, and results from the blank and control were deducted from that of the Fmoc-PXG/NO sample. The values of both the control and blank were found to remain roughly constant throughout the course of the experiment. Since it was a lengthy experiment, the values were also adjusted for solution evaporation. The nitrite released from the sample was quantified and plotted as a function of time (Figure 2).

The NO releasing profiles following resuspension in ultrapure water were also different. The 30 μM Fmoc-PXG functionalized sample released NO smoothly for a period that extended to over 15 days, with 50% of NO being released at around day three. This slow kinetics of NO release is quite promising for its application in wound dressing, as it allows for a continuous and slow release of the agent. However, the 400 μM Fmoc-PXG solution presented an undesired pronounced initial burst. Clearly the behavior of the functionalized peptide is very dependent on the reaction conditions used.

### 3.3. Antimicrobial Activity (Time-Kill Curve)

The quantification of bactericidal action of Fmoc-PXG and Fmoc-PXG/NO is represented in time-kill curves for different concentrations and summarized in Figure 3. These results are the product of arithmetic averages of duplicates.

For Fmoc-PXG/NO, bactericidal activity presumably arises from a sequence in time of nitric oxide released from the functionalized peptide into the culture media, followed by the well-known activity of the resulting Fmoc-PXG peptide. The time-kill curves suggest that Fmoc-PXG follows a consistent and slow antimicrobial profile. By contrast, Fmoc-PXG/NO presents a sharp initial antimicrobial action, an effect that is rapidly reversed in bacteria exposed to lower concentrations. This unexpected reduction in bactericidal activity, found in the later stages of the lower peptide concentration time-kill curves, can be related to some residual peptide degradation during the NO reaction, which would have affected the later part of the curves that reflect the action of the Fmoc-PXG peptide.

In the case of the control peptide (Fmoc-PXG) for concentrations above IC90 (19 μM and 37 μM), complete bacteria killing was confirmed at 5 h following incubation. However, in the case of Fmoc-PXG/NO, the time-point of complete killing was found to be concentration-dependent, with an accelerated action observed for higher concentrations. By increasing the concentration of Fmoc-PXG/NO two-fold, complete bacterial killing was verified in two hours rather than three. It should be mentioned that absolute killing values were restricted by the detection limit of the assay. This was determined as a function of the lower dilution of aliquot employed, which in the case of the points measured, was zero in a 100 µL aliquot. In accordance, the minimum CFU that was possible to quantify by the test was 10 CFU/mL.

Although the antimicrobial assays presented here do not characterize the complexity of an infected wound, these studies provide clues to the potential application of the newly developed Fmoc-PXG/NO.

### 3.4. In Vitro Assessment of Collagen Expression

Wound healing is a complex biological process that is initiated following tissue injury. The process involves a cascade of coordinated events that aim to restore both structural and functional integrity of damaged tissue. Different phases of wound healing may be recognized, including inflammation, proliferation and remodelling [56]. Collagen deposition by fibroblasts is particularly relevant within the proliferative phase when it replaces the provisional fibrin matrix, providing greater strength to the wound.

Here, we investigated whether Fmoc-PXG/NO contributes to an increase in collagen accumulation in fibroblasts. To that end, human dermal fibroblasts were cultured in the presence or absence of Fmoc-PXG/NO followed by the quantification of collagen deposition. The experimental design employed here was based on the work of Witte and colleagues, who studied the NO donor *S*-nitroso-*N*-acetyl-dl-penicillamine (SNAP) as an enhancer of collagen production [59]. In order to adjust their methods to our own experimental settings, SNAP was primarily used to replicate the published data using a different collagen quantification method. The accumulation of collagen has been quantitatively monitored by the colorimetric method of Sircol.

Fibroblast confluence was achieved at approximately 48 h following incubation. Microscopic examination, which was carried out approximately 23 h following the addition of the different components, showed no visible morphological changes at the concentrations considered. Collagen released into the culture media was quantified via Sircol assay and DNA was measured via Picogreen assay, as described previously. The results are depicted in Figure 4.

An initial assessment of the increasing amounts of collagen quantified from samples treated with progressively higher concentrations of NO donor suggests a positive correlation between collagen production associated with fibroblast exposure to NO donor.

Results of PicoGreen assays are shown in orange in Figure 4 and indicate that NO donor produces no significantly negative outcomes on cell number up to a concentration of 20 μM, above which the impact is quite notable. These results are consistent with microscopic observations, in which some cell detachment could be observed for concentrations above 50 μM. This was not unexpected since exposure of dermal fibroblasts to the NO donor SNAP at concentrations above 100 μM has been seen to result in significant decreases in the number of viable cells [60].

If considering the results of the Griess analysis, where a release of 168 μM of NO_2_^−^ per 100 μM Fmoc-PXG/NO was attained, a 50 μM Fmoc-PXG/NO sample would be expected to release around half of that value, 84 μM. This is in close proximity to the aforementioned threshold that other authors have attained for SNAP [60]. In a different study, in vitro cytotoxic tests of fibroblasts incubated with a NO-releasing zeolite revealed that only one third of the fibroblasts were viable after 24 h exposure to the NO-zeolite [61]. In this particular study, as far as we know, only one concentration was tested, thus precluding the evaluation of a threshold value. In conjunction with decreases in cell viability, a decrease in collagen was verified for concentrations above 50 μM (data not shown).

Peptide hydrolysis or uptake by cells may have direct relations to the significantly different results for the signals before and after its incubation with fibroblasts. Given the abrupt reduction in the signal and the lack of concentration dependency, it is reasonable to assume that the chromogenic precipitation reaction of Sirius Red in incubated samples is mostly a result of collagen production, and not peptide interference.

Acid hydrolysis followed by colorimetric hydroxyproline assays, immunoassays, and collagen mRNA quantification are among the most reliable and specific methods used for collagen quantification, and could be used as alternatives to Sircol assays. Collagen deposited in the extracellular matrix exhibits a profile similar to collagen released into the culture medium, as seen in Figure 5, with concentration dependence behavior. Accounting for the fact that the quantification of collagen deposited in the extracellular matrix (ECM) involves several washing steps, there is no interference of the peptide in the collagen quantification assay.

The larger standard deviation observed is most likely a consequence of the highly laborious procedure that was required to process these samples. Nevertheless, a statistically significant enhancement of collagen accumulation by fibroblasts, when incubated with Fmoc-PXG/NO, has been presented. In addition, the relationship between collagen and Fmoc-PXG/NO was found to be dose-dependent.

## 4. Conclusions

This study describes the production of an antimicrobial hydrogel formed by self-assembly of the peptide N-Fmoc-pexiganan, which is triggered by a pH shift. The peptide, which was subsequently reacted with gaseous NO to allow the incorporation of a NO donor moiety (NONOate), proved to release NO in aqueous conditions.

Time-kill curves were built to assess the antimicrobial activity of the NO-functionalized peptide and pointed to increased bactericidal activity in comparison with the unmodified peptide, which for low peptide concentrations saw reversed bactericidal activity with time. We believe that this effect may be a direct consequence of the release of NO, which is known to act as an antimicrobial agent. Optimization of reaction conditions may allow increases in the level of functionalization of the peptide, which could further raise the antimicrobial potential of Fmoc-PXG/NO.

Collagen production by human dermal fibroblasts incubated with Fmoc-PXG/NO was quantified, showing a dose-dependent increase in the presence of NO donor within a range of 0–20 μM. Although additional experiments are still required in order to obtain a hydrogel with optimized antimicrobial activity and wound healing properties, for biomedical applications, the present work constitutes an essential step towards that end as its findings are strongly encouraging.

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
