# Peer review of "Nitric Oxide Release from Antimicrobial Peptide Hydrogels for Wound Healing"

_biomolecules, 2018, doi:10.3390/biom9010004_

Round 1
Reviewer 1 Report
Gales and coworkers reported the fabrication of NO releasing from antimicrobial peptide hydrogels and its application in wound healing. The results are encouraging and I suggest the publication of this manuscript after minor revision. See more comments below.
1. The authors are encouraged to include more data on the cytotoxicity of the NO releasing hydrogels, which is important for wound healing application.
2. Some key references for nitric oxide release, and its synergy with antimicrobial peptide, and its application in wound healing is missing. For example:
a) Ren, H.; Colletta, A.; Koley, D.; Wu, J.; Xi, C.; Major, T. C.; Bartlett, R. H.; Meyerhoff, M. E., Thromboresistant/anti-biofilm catheters via electrochemically modulated nitric oxide release. Bioelectrochemistry 2015, 104 (0), 10-16.
b) Ren, H.; Wu, J. F.; Colletta, A.; Meyerhoff, M. E.; Xi, C. W., Efficient Eradication of Mature Pseudomonas aeruginosa Biofilm via Controlled Delivery of Nitric Oxide Combined with Antimicrobial Peptide and Antibiotics. Frontiers in Microbiology 2016, 7.
c) Lee, W. H.; Ren, H.; Wu, J.; Novak, O.; Brown, R. B.; Xi, C.; Meyerhoff, M. E., Electrochemically Modulated Nitric Oxide Release From Flexible Silicone Rubber Patch: Antimicrobial Activity For Potential Wound Healing Applications. Acs Biomater Sci Eng 2016, 2 (9), 1432-1435.
Author Response
We thank the reviewer for the observations; we also acknowledge the time spent in the review and the perceptive comments that will undoubtedly improve the quality of the manuscript. The response is addressed below.
Reviewer 1:
Gales and coworkers reported the fabrication of NO releasing from antimicrobial peptide hydrogels and its application in wound healing. The results are encouraging and I suggest the publication of this manuscript after minor revision. See more comments below.
1. The authors are encouraged to include more data on the cytotoxicity of the NO releasing hydrogels, which is important for wound healing application.
Yes, we agree with the reviewer. We believed that we should publish immediately our initial promising results but we have started with new experiments to obtain further investigation on the cytotoxicity of the NO releasing hydrogels.
2. Some key references for nitric oxide release, and its synergy with antimicrobial peptide, and its application in wound healing is missing. For example:
a) Ren, H.; Colletta, A.; Koley, D.; Wu, J.; Xi, C.; Major, T. C.; Bartlett, R. H.; Meyerhoff, M. E., Thromboresistant/anti-biofilm catheters via electrochemically modulated nitric oxide release. Bioelectrochemistry 2015, 104 (0), 10-16.
b) Ren, H.; Wu, J. F.; Colletta, A.; Meyerhoff, M. E.; Xi, C. W., Efficient Eradication of Mature Pseudomonas aeruginosa Biofilm via Controlled Delivery of Nitric Oxide Combined with Antimicrobial Peptide and Antibiotics. Frontiers in Microbiology 2016, 7.
c) Lee, W. H.; Ren, H.; Wu, J.; Novak, O.; Brown, R. B.; Xi, C.; Meyerhoff, M. E., Electrochemically Modulated Nitric Oxide Release From Flexible Silicone Rubber Patch: Antimicrobial Activity For Potential Wound Healing Applications. Acs Biomater Sci Eng 2016, 2 (9), 1432-1435.
Yes, we agree. We thank the reviewer for the observations. Indeed, these new references were introduced in the manuscript, as:
[8] Ren, H.; Colletta, A.; Koley, D.; Wu, J.; Xi, C.; Major, T. C.; Bartlett, R. H.; Meyerhoff, M. E., Thromboresistant/anti-biofilm catheters via electrochemically modulated nitric oxide release. Bioelectrochemistry 2015, 104 (0), 10-16 DOI: 10.1016/j.bioelechem.2014.12.003
[9] Ren, H.; Wu, J. F.; Colletta, A.; Meyerhoff, M. E.; Xi, C. W., Efficient eradication of mature pseudomonas aeruginosa biofilm via controlled delivery of nitric oxide combined with antimicrobial peptide and antibiotics. Front. Microbiol. 2016, 7, Article 1260 DOI: 10.3389/fmicb.2016.01260
[10] Lee, W. H.; Ren, H.; Wu, J.; Novak, O.; Brown, R. B.; Xi, C.; Meyerhoff, M. E., Electrochemically modulated nitric oxide release from flexible silicone rubber patch: antimicrobial activity for potential wound healing applications. Acs Biomater. Sci. Eng. 2016, 2 (9), 1432-1435 DOI: 10.1021/acsbiomaterials.6b00360
Reviewer 2 Report
The manuscript by DurĂŁo et al. describes the synthesis and characterization of nitric oxide (NO) as an antimicrobial peptide. Upon synthesis with a pH shift, NO self-assembled in a hydrogel. The reaction was measured by the ninhydrin assay and the NO-release rate was quantified via Griess reaction method. The authors assessed the NO compound for its antimicrobial activity in assays with Escherichia coli and for its effect on collagen production by fibroblasts. The authors have supported their claims with clearly defined experiments and the manuscript is written well with apt background information. This study is interesting; however, the authors should add some experiments to confirm their findings.
Major revisions:
1) The authors present all analyses only as a graph. It would be interesting and essential for the manuscript to also add some pictures for example to show fibroblast attachment, cell viability, and proliferation assays in Fmoc-PXG/NO versus control gels.
2) Figure 4-5: collagen accumulation in medium and its deposition in gels: What was used as a negative control for those experiments? Why is it not shown in the results? Please explain.
3) What the authors mean with “SNAP”? Please explain it in materials and methods. It is not clear from the text.
4) Why there are increasing amounts of collagen quantified from samples treated with progressively higher concentrations of NO? Please explain and add some references if available.
Author Response
We thank the reviewer for the observations; we also acknowledge the time spent in the review and the perceptive comments that will undoubtedly improve the quality of the manuscript. The response is addressed below.
Reviewer 2:
The manuscript by DurĂŁo et al. describes the synthesis and characterization of nitric oxide (NO) as an antimicrobial peptide. Upon synthesis with a pH shift, NO self-assembled in a hydrogel. The reaction was measured by the ninhydrin assay and the NO-release rate was quantified via Griess reaction method. The authors assessed the NO compound for its antimicrobial activity in assays with Escherichia coli and for its effect on collagen production by fibroblasts. The authors have supported their claims with clearly defined experiments and the manuscript is written well with apt background information. This study is interesting; however, the authors should add some experiments to confirm their findings.
Major revisions:
1) The authors present all analyses only as a graph. It would be interesting and essential for the manuscript to also add some pictures for example to show fibroblast attachment, cell viability, and proliferation assays in Fmoc-PXG/NO versus control gels.
We do agree with this observation on present some pictures to fibroblast attachment, cell viability and proliferation assays; however, we believed that we should publish immediately our initial promising results. Recently, we have started with new experiments to obtain further investigation using Fmoc-PXG/NO vs control gels.
2. Figure 4-5: collagen accumulation in medium and its deposition in gels: What was used as a negative control for those experiments? Why is it not shown in the results? Please explain.
Thank you for the observation. Our experiments was supported by work from Lareu et al (Essential modification of the Sircol Collagen Assay for the accurate quantification
of collagen content in complex protein solutions, Acta Biomaterialia 2010, 6, 3146-3151). The negative control was solvent.
3. What the authors mean with “SNAP”? Please explain it in materials and methods. It is not clear from the text.
Yes, we agree. SNAP is the compound S-Nitroso-N-acetyl-DL-penicillamine (synonym: N-Acetyl-3-(nitrosothio)-DL-valine, S-Nitroso-N-acetylpenicillamine), with CAS Number 67776-06-1. SNAP is an S-nitrosothiol and is used as a model for the general class of S-nitrosothiols which have received much attention in biochemistry because nitric oxide and some organic nitroso derivatives serve as signaling molecules in living systems, especially related to vasodilation.
4. Why there are increasing amounts of collagen quantified from samples treated with progressively higher concentrations of NO? Please explain and add some references if available.
Fibroblast confluence was achieved at approximately 48 hours following incubation. Microscopic examination, approximately 23 hours following the addition of the different components, showed no visible morphological changes at the concentrations hereby considered. Collagen released into the culture media was quantified via Sircol assay and DNA measured through Picogreen assay, as described previously. Results are depicted on next Figure (Collagen accumulated in the culture medium as a function of NO donor concentration)
Collagen is graphed in blue columns and DNA in orange triangles at each concentration. Standard deviations are represented in bars and results from three independent experiments.
A primary appraisal of the increasing amounts of collagen quantified from samples treated with progressively higher concentrations of NO donor, suggests a positive correlation between collagen production associated with fibroblast exposure to NO donor.
Results of PicoGreen assay are shown in orange in Figure 1 and indicate that NO donor produces no significantly negative outcome on cell number, up to a concentration of 20 mM, above which the impact is quite expressive (data not shown). These results are consistent with microscopic observations, in which some cell detachment can be observed for concentrations above 50 mM. This was not unexpected since exposure of dermal fibroblasts to the NO donor SNAP, at concentrations above 100 mM, resulted in the significant decrease on the number of viable cells (Witte, M.B., et al., Enhancement of Fibroblast Collagen Synthesis by Nitric Oxide. Nitric Oxide, 2000. 4(6): p. 572-582).
Considering the results of the Griess analysis previously presented, wherein a release of 168 mM of NO2- per 100 mM Fmoc-PXG/NO was attained, then accordingly, a 50 mM Fmoc-PXG/NO sample is expected to release around half of that value, 84 mM. This is in close proximity to the abovementioned threshold that other authors attained for SNAP (Witte, M.B., et al., Enhancement of Fibroblast Collagen Synthesis by Nitric Oxide. Nitric Oxide, 2000. 4(6): p. 572-582). In a different study, in vitro cytotoxic tests of fibroblasts incubated with an NO-releasing zeolite, revealed that only one third of the fibroblasts were viable after a 24h exposure to the NO-zeolite (Neidrauer, M., et al., Antimicrobial efficacy and wound-healing property of a topical ointment containing nitric-oxide-loaded zeolites. Journal of Medical Microbiology, 2014. 63(2): p. 203-209). In this particular study, as far as we know, only one concentration was tested, thus, precluding the evaluation of a threshold value.
Round 2
Reviewer 2 Report
Thank you for the authors' reponse, hovewer I still do not see any changes in the manuscript regarding my questions, for example Comment #3: Where in the text the authors explain properly "SNAP"?
Author Response
Reviewer 2:
Thank you for the authors' reponse, hovewer I still do not see any changes in the manuscript regarding my questions, for example Comment #3: Where in the text the authors explain properly "SNAP"?
We thank again the Reviewer for the perceptive comments and observations.
You are right. This information was mentioned in Points 2 and 3 but missed in the manuscript. Apologies for this.
About SNAP: now, we can read on lines 312-314
“The experimental design here employed was based on the work by Witte and colleges, who studied the NO donor S-nitroso-N-acetyl-DL-penicillamine (SNAP) as an enhancer of collagen production [59].”
(first explanation)
“What the authors mean with “SNAP”? Please explain it in materials and methods. It is not clear from the text.”
Yes, we agree. SNAP is the compound S-Nitroso-N-acetyl-DL-penicillamine (synonym: N-Acetyl-3-(nitrosothio)-DL-valine, S-Nitroso-N-acetylpenicillamine), with CAS Number 67776-06-1. SNAP is an S-nitrosothiol and is used as a model for the general class of S-nitrosothiols which have received much attention in biochemistry because nitric oxide and some organic nitroso derivatives serve as signaling molecules in living systems, especially related to vasodilation.
About solvent: now, we can read on lines 207-209
“Collagen was assessed by Sircol assay (Biocolor) according to instructions provided by the manufacturer, with the exception of the Isolation and Concentration step, which was replaced by an improved procedure [57], and the negative control was solvent.”
(first explanation)
Figure 4-5: collagen accumulation in medium and its deposition in gels: What was used as a negative control for those experiments? Why is it not shown in the results? Please explain.
Thank you for the observation. Our experiments was supported by work from Lareu et al (Essential modification of the Sircol Collagen Assay for the accurate quantification
of collagen content in complex protein solutions, Acta Biomaterialia 2010, 6, 3146-3151). The negative control was solvent.